# Ultrabright Fluorescent Silica Nanoparticles for Multiplexed Detection

**DOI:** 10.3390/nano10050905

**Published:** 2020-05-08

**Authors:** Saquib Ahmed M. A. Peerzade, Nadezda Makarova, Igor Sokolov

**Affiliations:** 1Department of Biomedical Engineering, Tufts University, Medford, MA 02155, USA; Saquib.Peerzade@tufts.edu; 2Department of Mechanical Engineering, Tufts University, Medford, MA 02155, USA; Nadezda.Makarova@tufts.edu; 3Department of Physics, Tufts University, Medford, MA 02155, USA

**Keywords:** fluorescent nanoparticles, fluorescence, multiplexing

## Abstract

Fluorescent tagging is a popular method in biomedical research. Using multiple taggants of different but resolvable fluorescent spectra simultaneously (multiplexing), it is possible to obtain more comprehensive and faster information about various biochemical reactions and diseases, for example, in the method of flow cytometry. Here we report on a first demonstration of the synthesis of ultrabright fluorescent silica nanoporous nanoparticles (Star-dots), which have a large number of complex fluorescence spectra suitable for multiplexed applications. The spectra are obtained via simple physical mixing of different commercially available fluorescent dyes in a synthesizing bath. The resulting particles contain dye molecules encapsulated inside of cylindrical nanochannels of the silica matrix. The distance between the dye molecules is sufficiently small to attain Forster resonance energy transfer (FRET) coupling within a portion of the encapsulated dye molecules. As a result, one can have particles of multiple spectra that can be excited with just one wavelength. We show this for the mixing of five, three, and two dyes. Furthermore, the dyes can be mixed inside of particles in different proportions. This brings another dimension in the complexity of the obtained spectra and makes the number of different resolvable spectra practically unlimited. We demonstrate that the spectra obtained by different mixing of just two dyes inside of each particle can be easily distinguished by using a linear decomposition method. As a practical example, the errors of demultiplexing are measured when sets of a hundred particles are used for tagging.

## 1. Introduction

Optical microscopy and detection methods, which employ a wide variety of non-invasive optical techniques, are fundamental to understanding the structural, organizational, and dynamic properties of biological systems. Among those optical detection modalities, fluorescence microscopy and flow cytometry are particularly important because they facilitate highly sensitive and specific imaging and detection. Fluorescent nanoparticles are becoming increasingly popular in biomedical imaging and sensing. When functionalized with tagging molecules, such nanoparticles can be used for the detection and multiplexed imaging of specific (large) molecules, cells, and tissues [1,2,3,4,5,6]. A particular interest in such particles is in flow cytometry, a method that counts the number of fluorescent markers per an object of investigation, for example, cells, exosomes, and liposomes.

Several groups are currently trying to develop such particles [7,8,9,10,11,12,13]. The most developed (and commercially available) fluorescent nanoparticles are quantum dots (QDs) [12,13,14,15]. Although excellent in many aspects, they still exhibit several problems, including potential toxicity, some fractions of non-fluorescent QDs, blinking, etc. [16]. Polymer dots [11,17,18], an organic version of QDs, are promising bright particles that are still under development and have a limited number of available semiconductor polymers. Biodegradation, long-term toxicity, and monodispersity of polymer dots still have to be investigated/improved. Solid dye-doped silica particles were used for a while [5] which are not that bright and require chemical modification of dye, which limits the number of available dyes and adds to the cost.

Ultrabright fluorescent nanoporous silica nanoparticles (Star-dots) are a new class of exceptionally bright fluorescent silica nanoparticles [19,20,21]. The brightness of such particles is substantially higher than the fluorescence of individual dye molecules (for example, the brightness of a 30 nm particle can be equivalent to the fluorescent brightness of 600–2000 encapsulated dye molecules [22]). This makes them one of the brightest (if not the brightest) fluorescent nanoparticles currently available. The use of Star-dots will substantially expand imaging and diagnosis involving fluorescence. They have already shown compatibility with biological cells and in vivo applications [23,24].

The ultimate fluorescent brightness of Star-dots comes from the special nano environment of the encapsulated dye molecules. As was demonstrated in [22], the encapsulated dye molecules are encaged inside of cylindrical nanochannels, and the collisional quenching of fluorescence of the encapsulated dye molecules is de facto eliminated. At the same time, each dye molecule is virtually levitating in the environment created by alkane chains of the templating surfactant molecules. Thus, the dye molecules can be packed very close to each other without any decrease in their quantum yield. As a result, the particles possess an extremely bright fluorescence. So far, the synthesis of Star dots has been shown only for a single dye.

In this work, we demonstrate for the first time that the synthesis of Star-dots can be modified to encapsulate multiple dyes. Furthermore, due to the close proximity of the encapsulated dye molecules to each other, it is possible to engage the Forster resonance energy transfer (FRET) between different encapsulated dye molecules. This can create Star-dots with a large number of different fluorescent spectra that can be excited with the same excitation wavelength. The latter is preferable for the technical implementation of the detection of multiplexed fluorescence in a quantitative manner.

Besides the fundamental interest in creating new photonic nanomaterials, the created nanoparticles can be used in multiple biomedical applications for the identification of particular molecules, diseases, and cell receptors, in which multiplexing is advantageous. Besides multiplex imaging, the particles can be directly used in such a popular diagnostic method as flow cytometry. The high brightness of the particles is advantageous, for example, in flow cytometry because it allows cells with a small number of receptors to be identified, extending flow cytometry to extracellular vesicles [25]. Furthermore, high brightness allows the use of less excitation light, which results in less phototoxicity.

Here we report on the synthesis of Star-dots in which complex fluorescent spectra are obtained by simple physical mixing of different commercially available fluorescent dyes in the synthesizing bath. It is demonstrated for the mixing of five, three, and two different dyes, which are in part FRET-coupled. In addition, we demonstrate the encapsulation of dyes mixed inside of particles in different proportions. This brings another dimension in the complexity of the obtained spectrum. We demonstrate that the spectra obtained by mixing just two different dyes can be easily distinguished by using a linear decomposition method. In combination with multiple dyes, it demonstrates that the previously limited number of fluorescent spectra may be no longer a limitation in any multiplexed applications when using Star-dots.

## 2. Materials and Methods

### 2.1. Materials

Tetraethylorthosilicate (TEOS, ≥99%, GC, Acros Organics, Fair Lawn, NJ, USA), triethanolamine (TEA, reagent grade 98%, Sigma Aldrich, St. Louis, MO, USA), cetyltrimethylammonium bromide (CTAB, High Purity Grade, Amresco, Solon, OH, USA), ethyltriethoxysilane (ETES, 96%, Frontier Scientific, Logan, UT, USA), rhodamine 6G (R6G, Sigma Aldrich, St. Louis, MO, USA), coumarin 504 (C504, Exciton, Dayton, OH, USA), rhodamine 560 (R560, Exciton, Dayton, OH, USA), rhodamine 640 (R640, Exciton, Dayton, OH, USA), fluorescein isothiocyanate (FITC, Exciton, Dayton, OH, USA), rhodamine B (RB, Exciton, Dayton, OH, USA), and Nile blue A (NB, Electron Microscopy Sciences, Hatfield, PA, USA) were used. RC membrane (RC membrane, Spectra/Pore, Rancho Daminguez, CA, USA) with 10–15 kDa MW was used. Deionized water was used for all synthesis.

### 2.2. Particle Synthesis

**Synthesis of Star-dots with five encapsulated dyes (Silica multi-color (Si-MC))**: A previously reported procedure was modified for making Star dots encapsulated with five dyes based on mesoporous silica particles [24,26]. The molar ratio was 1 TEOS:8.2 TEA:0.23 CTAB:142 H_2_O:0.1 ETES. The ratio of C504:R560:R6G:NB was 1:1:2:1:3:3. The mixture of TEOS (1.71 g, 8.2 mmol) and TEA (10 g, 67 mmol) was stirred for one minute and kept at 90 °C under quiescent conditions for 20 min. Another mixture of CTAB (0.69 g, 1.9 mmol), C504 (0.005 g, 0.016 mmol), R560 (0.008 g, 0.016 mmol), R6G (0.015 g, 0.032 mmol), R640 (0.009 g, 0.016 mmol), and NB (0.017 g, 0.048 mmol) was dissolved in ethanol (3 mL) and H_2_O (21 mL) and kept at 60 °C for 40 min. The CTAB, dye, and water mixture were allowed to stir at room temperature for another 15 min and was kept in a cold bath for 5 min. After 5 min the mixture of TEOS and TEA was then added to the aqueous solution of CTAB and dye. ETES (130 µL, 0.8 mmol) was added after 10–15 min and was stirred for a further 40 min in the cold bath. After 40 min, the synthesis mixture was diluted with 30 mL water, and the excess reagents were removed by dialyzing with water using the membrane of MW 10–15 kDa until no fluorescence was obtained from the dialysate (several (2–3) days). The pH of the mixture after dialysis was ~9. HCl was added to reduce the pH to 7.

**Synthesis of Star-dots with three encapsulated dyes (Si-FITC-RB-NB)**: A previously reported procedure was modified for making Star dots encapsulated with three dyes based on mesoporous silica particles [24]. The molar ratio was 1 TEOS:8.2 TEA:0.23 CTAB:142 H_2_O:0.1 ETES. The ratio of FITC:RB:NB was 1:4:55. The mixture of TEOS (1.71 g, 8.2 mmol) and TEA (10 g, 67 mmol) was stirred for one minute and kept at 90 °C under quiescent conditions for 20 min. Another mixture of CTAB (0.69 g, 1.9 mmol), FITC (0.001 g, 0.0026 mmol), RB (0.005 g, 0.013 mmol), and NB (0.05 g, 0.14 mmol) was dissolved in H_2_O (21 mL) and kept at 60 °C for 40 min. The CTAB, dye, and water mixture were allowed to stir at room temperature for another 15 min and was kept in a cold bath for 5 min. After 5 min, the mixture of TEOS and TEA was then added to the aqueous solution of CTAB and dye. ETES (130 µL, 0.8 mmol) was added after 10–15 min and was stirred for a further 40 min in the cold bath. After 40 min, the synthesis mixture was diluted with 30 mL water, and the excess reagents were removed by dialyzing with water using the membrane of MW 10–15 kDa until no fluorescence was obtained from the dialysate (several (2–3) days). The pH of the mixture after dialysis was ~9. HCl was added to reduce the pH to 7.

**Synthesis of Star-dots with two encapsulated dyes**: A previously reported procedure was modified for making Star dots encapsulated with two dyes based on mesoporous silica particles [21,23,24]. The molar ratio was 1 TEOS:11.7 TEA:0.23 CTAB:142 H_2_O:0.11 ETES. The molar ratio (MR) of R6G:RB was 1:0.1 (MR1:0.1) and 1:1 (MR1:1). The mixture of TEOS (1.71 g, 8.2 mmol) and TEA (14.3 g, 96 mmol) was stirred for one minute and kept at 90 °C under quiescent conditions for 3 h. Another mixture of CTAB (0.69 g, 1.9 mmol), R6G (0.073 g, 0.15 mmol for all molar ratios), and RB ((0.0073 g, 0.015 mmol) for MR1:0.1 and (0.073 g, 0.15 mmol) for MR1:1) was dissolved in H_2_O (21 mL) and kept at 60 °C for 40 min. The CTAB, dye, and water mixture were allowed to stir at room temperature for another 15 min. The mixture of TEOS and TEA was then added to the aqueous solution of CTAB and dye. ETES (196 µL, 0.9 mmol) was added after 30 min and was stirred for further 3 h. The excess reagents were removed by dialyzing with water using the membrane of MW 10–15 kDa until no fluorescence was obtained from the dialysate.

### 2.3. Characterization Techniques

The particle size was measured with the help of the dynamic light scattering (DLS) technique (Zetasizer Nano ZS by Malvern Instruments Ltd., Malvern, UK). The Z-average size and most probable size (mean of number weighted distribution) was measured at least three times. DLS uses the laser light of 633 nm and the backscattered light is monitored over light at an angle of 173 °C; 0.1 mL of stock solution was diluted to 3 mL deionized water before measurements. Particles were weighed for determining particle concentration. Triplicates of 0.1 mL of a water suspension of particles in an aluminum foil cap were dried in a vacuum chamber for 24 h and weighed using a CAHN29 (CAHN Instruments Inc., Paramount, CA, USA). A Cary 60 UV–Vis spectrometer from Agilent technologies was used to measure absorbance. Fluorescence was measured using the Cary Eclipse (Agilent, Santa Clara, CA, USA) and Horiba Fluorelog 3 (Horiba, Japan). Fluorescence spectra of a small set of nanoparticles were recorded using a confocal Raman microscope Alpha 300 (WITec, Inc., Ulm, Germany). An Icon Atomic Force Microscopy (AFM, Bruker, Inc., Santa Barbara, CA, USA) with NanoScope V controller with Ringing mode add-on (NanoScience Solutions, Inc., Arlington, VA, USA) was used to image the obtained nanoparticles.

### 2.4. Linear Decomposition of Complex Spectra

A multiplexing approach means the ability to identify concentrations of individual fluorescent markers using their individual spectra. Due to a relatively broad spectrum of fluorescence of organic dyes, the multiplexing spectra of particles encapsulating several dyes have overlapping emission. Nonetheless, it is possible to identify the contributions of individual fluorophores despite this overlap. The linear spectral decomposition is a traditional approach to calculate the contributions of each fluorophore in the mixed fluorescence spectrum independently of the overlap. In this approach, the measured spectrum is treated as a sum of the spectra of each component *i* weighted with its concentration $c_{i}$ [27,28].
(1)$$FL_{T}^{\lambda^{1}}=\sum_{i=1}^{n}c_{i}FL_{i}^{\lambda^{1}},$$
where $FL_{T}^{\lambda^{1}}\;$ is the total measured intensity at the wavelength $\lambda^{1}$ for the mixed spectra; and $FL_{i}^{\lambda^{1}}\;$ is the spectral intensity of the *it*h-component at the wavelength $\lambda^{1}$.

Considering this as a system of linear equations, one can see that the system is overdetermined (it has more data than unknown parameters $c_{i}$). Furthermore, the measured parameters have some experimental measurement errors. Thus, the values of the concentrations can be found using the standard minimization of the residuals of the difference between the observed and calculated spectra. The residuals can be found using the following equation:(2)$$S=\overset{m}{\underset{i=1}{\sum }}{\left[FL_{T}^{\lambda^{i}}-\overset{n}{\underset{j=1}{\sum }}c_{j}FL_{j}^{\lambda^{i}}\right]}^{2}\;$$

To find the minimum residual, one needs to zero the derivative of the residual with respect to the unknown concentrations. As an example, the obtained system of linear equations for the case of two fluorophores (n = 2), is written as
(3)$$\overset{m}{\underset{i=1}{\sum }}FL_{T}^{\lambda^{i}}FL_{a}^{\lambda^{i}}-c_{a}\overset{m}{\underset{i=1}{\sum }}FL_{a}^{\lambda^{i}}FL_{a}^{\lambda^{i}}-c_{b}\overset{m}{\underset{i=1}{\sum }}FL_{a}^{\lambda^{i}}FL_{b}^{\lambda^{i}}=0$$
(4)$$\overset{m}{\underset{i=1}{\sum }}FL_{T}^{\lambda^{i}}FL_{b}^{\lambda^{i}}-c_{b}\overset{m}{\underset{i=1}{\sum }}FL_{b}^{\lambda^{i}}FL_{b}^{\lambda^{i}}-c_{a}\overset{m}{\underset{i=1}{\sum }}FL_{a}^{\lambda^{i}}FL_{b}^{\lambda^{i}}=0\;$$
where indices a and b stand for two different fluorophores.

The sought concentrations, *c_i_*, can be found by solving this system of linear equations. For simplicity, it can be rewritten in the matrix form as follows:(5)$$\left[\begin{array}{cc} FL_{aa} & FL_{ab}\\ FL_{ab} & FL_{bb} \end{array}\right]\;\left[\begin{array}{c} c_{a}\\ c_{b} \end{array}\right]=\left[\begin{array}{c} FL_{Ta}\\ FL_{Tb} \end{array}\right]$$
where $FL_{xy}=\sum_{i=1}^{n}FL_{x}^{\lambda^{i}}FL_{y}^{\lambda^{i}}$.

Thus, the concentrations can be found using the following equation:(6)$$\left[\begin{array}{c} c_{a}\\ c_{b} \end{array}\right]={\left[\begin{array}{cc} FL_{aa} & FL_{ab}\\ FL_{ab} & FL_{bb} \end{array}\right]}^{-1}\;\left[\begin{array}{c} FL_{Ta}\\ FL_{Tb} \end{array}\right]=\frac{1}{FL_{aa}FL_{bb}-FL_{ab}{}^{2}}\left[\begin{array}{cc} FL_{bb} & -FL_{ab}\\ -FL_{ab} & FL_{aa} \end{array}\right]\;\left[\begin{array}{c} FL_{Ta}\\ FL_{Tb} \end{array}\right]$$

It should be noted that if overlapping spectra are similar (ratio between the spectra are close to a constant), the problem of finding the concentrations is mathematically ill-defined. Specifically, in terms of Equation (6), the denominator $FL_{aa}FL_{bb}-FL_{ab}{}^{2}\;$ tends to zero. In general, this denominator is the determinant of the fluorescence matrix, an example of which is given in Equation (5) for two fluorophores. Thus, this determinant can be considered as an estimate of how good the method of linear decomposition can be.

## 3. Results and Discussion

Here we use a templated sol-gel chemistry self-assembly mechanism [22,29,30], which was adapted to synthesize Star-dots with a single dye [19,21]. The nanostructure and porosity of this material were presented in previous publications [19,21]. Individual dye molecules are encapsulated inside of nanoscopic cylindrical channels formed by the nematic phase of the liquid crystal, which in turn is created by the templating surfactant molecules. The inner space of the cylindrical channels is filled with alkane chains of surfactant (CTAB in our case).

Below, we present the results of the syntheses of Star-dots, which encapsulated five (C504, R560, R6G, R640, and NB), three (FITC, RB, and NB), and two (R6G and RB) different fluorescent dyes. The dye molecules were physically coupled by FRET to transfer their excitation energy from the donor to the acceptor dye. The nature of a particular encapsulated dye showed little influence on the nanostructure of the particles [31]. Figure 1 shows the particle size distribution obtained with the DLS technique (see also Appendix A). The inserts show the AFM images of the particles dispersed on a glass slide. The AFM images are intended to show the mostly roundish geometry of the particles of approximately the same size as measured by DLS. One can see that the particle size was very weakly dependent on the encapsulated dyes. Furthermore, based on the FRET observed here and high quantum yield (see below), we can conclude that the use of multiple dyes for encapsulation did not noticeably influence the geometry of Star-dots.

Figure 2 shows the case of encapsulation of five dyes inside of Star-dots: C504, R560, R6G, R640, and NB. Figure 2a shows the absorption and emission spectra of each dye. The overlap between the emission of donor and absorbance of acceptor dyes, which are clearly seen in this figure, yielded a FRET cascade in fluorescence excitation/emission within each particle. Figure 2b demonstrates a fluorescence spectrum in which the excitation wavelength was moved between 400 and 700 nm, while the emission was measured at the wavelengths of the excitation plus 10 nm (synchronized spectrum). In this way, one could clearly see the presence of individual dye molecules encapsulated in particles. Finally, Figure 2c shows the fluorescence spectrum of Star-dots excited at 480 nm. The Star-dots yielded a fluorescence spectrum spanning from 470 to 700 nm. The decomposition of the obtained spectrum onto the contribution from individual dyes (fluorescence spectrum was recorded for water solution of dyes) is also shown. One can see that the linear mixing of the contributed dyes matched rather well with the measured fluorescence spectrum.

The observed peak positions were slightly blue-shifted for several dyes relative to their emission in water. The first donor dye was C504, which showed the maximum absorption at 450 nm and emission at 500 nm. The emission maximum was 490 nm for C504 (490 nm in water), 521 nm for R560 (521 nm in water), 547 nm for R6G (552 nm in water), 597 nm for R640 (597 nm in water), and 629 nm for NB (677 nm in water). The quantum yields of the dyes used in this work are presented in Appendix A. The emission from the last dye in FRET order, NB, was rather low due to the low concentration in the particles, as seen from the comparison of the synchronized and fluorescence spectra. The observed blue shift of R6G and NB dyes was in a good agreement with the previous reports of the encapsulation of single dyes [19,21,22,31]. It was explained by interactions of the dye molecules with the low polarity environment created by the alkane chains of the templating surfactant molecules.

Comparing the spectra in Figure 2, one can clearly see the presence of FRET. For example, the NB fluorescence peak was seen in the fluorescence spectra of the particles (Figure 2c), while the fluorescence was being excited with light at 480 nm. Looking at the absorption spectra of NB dye at that wavelength, one could see that no fluorescence should have been excited if there were no FRET. Furthermore, one could see that the amount of fluorescence activity was increasing for R560, R6G, and R640 dyes by looking at the synchronized spectrum (when excited at the corresponding dye maximum absorbance). For the sake of simplicity, we demonstrate FRET quantitatively for the case of two dyes later in this paper.

Figure 3 shows the case of encapsulation of three dyes inside of Star-dots, FITC, RB, and NB dyes. Figure 3a shows the absorption and emission spectra of each dye used for encapsulation in these Star-dots. One can clearly see the overlap between the emission of donor and absorbance of acceptor dyes, which yielded the FRET cascade in fluorescence excitation/emission within each particle. Figure 3b demonstrates a fluorescence spectrum in which excitation was scanned between 400 and 700 nm, and the emission was measured at the wavelengths of the excitation plus 10 nm (synchronized spectrum). One can clearly see the presence of individual dye molecules encapsulated in particles, although the presence of FITC was almost hidden due to the efficient FRET coupling with RB dye. This coupling can be seen in Figure 3c, which shows the fluorescence spectrum of Star-dots excited at 488 nm. One can see that virtually all FITC absorbed energy was transferred to the excitation of RB dye, which in turn, shared its excitation with NB dye.

These Star-dots yielded a fluorescence spectrum spanning from 530 to 700 nm. A linear decomposition of the obtained spectrum onto the contribution from individual dyes (in water solution) is also shown. The presence of FRET is clearly seen in the fluorescence spectrum of the particles shown in Figure 3c. Specifically, the part of the fluorescence spectrum coming from NB was pretty significant despite NB dye having practically zero absorbance at 488 nm.

Now we demonstrated an example of encapsulation of two different dyes, R6G and RB, which were encapsulated inside of the particles in two different proportions. Absorption/fluorescence spectra of these dyes are already shown in Figure 2 and Figure 3. Figure 4 shows the fluorescent spectra of two different proportions of these dyes with molar ratios of R6G:RB of 1:1 and 1:0.1. The fluorescence spectra were excited with a wavelength of 488 nm. One can see the spectra were entirely overlapping. Nevertheless, as we show later, these spectra could be demultiplexed with a rather small error.

Let us demonstrate that the observed spectra of these 2-dye particles were in agreement with the presence of FRET coupling between a portion of the encapsulated dye molecules. To estimate FRET, one needs to know the number of the dye molecules of each type inside of particles. Thus, we first found the number of encapsulated fluorescent molecules inside the particles. We then also were able to calculate the absolute brightness of the particles.

Table 1 shows the calculation of the number of molecules per particle. It was done as described in [19,21,22,31]. Briefly, the concentration of each dye in the suspension of nanoparticle was measured using the known absorptivity for each dye. Note that the method takes into account the scattering of light by the particles. The number of the particles in the suspension is estimated by drying and weighting of the physical mass of the particles. Assuming all particles of the same diameter (Z average is defined by the DLS technique), one can find the average number of the dye molecules of each type encapsulated inside of each particle.

To calculate the fluorescence brightness of the particles, the MESF units (molecules of equivalent soluble fluorochrome) are frequently used [14,26,31,32,33,34,35]. These units are popular in applications in which the quantitative nature of measurements is important, such as flow cytometry. It is explained by the unambiguous definition of the brightness, which is defined as follows:(7)$$\mathrm{Brightness}\;(\mathrm{in}\;\mathrm{MESF}\;\mathrm{units})=\frac{FL_{p}}{C_{p}}/\frac{FL_{d}}{C_{d}},$$
where *FL* refers to the integral fluorescence, and *C* is the concentration. The subscript *p* indicates particles and *d* is for a reference dye.

The above equation is typically used with a reference dye. Star-dots are the particles with complex spectra, containing multiple dyes. Thus, it is plausible to define the brightness with respect to the fluorescence of each encapsulated dye. Furthermore, each reference dye can potentially be excited either with the same excitation light as the particles of interest or at its maximum absorbance of the reference dye. This ambiguity in the description of the brightness of particles with a complex fluorescent spectrum is inevitable. In the present case, R6G and RB dyes are used as the reference dyes. To find the brightness, the Star-dot spectrum was decomposed to R6G and RB individual components by using the linear decomposition as described in the Methods section (the graphical results of the decomposition are shown in Appendix A). The brightness of each component was calculated with respect to the fluorescent emission of each reference dye excited at 488 nm (the same as Star-dots) and with respect to the absorbance maximum of each reference dye (550 nm for RB and 525 nm for R6G). These results are also presented in Table 1.

One can now see the presence of FRET from Table 1. For example, based on the number of encapsulated dye molecules, each particle should produce fluorescence equal to the one coming from 226 molecules of R6G and 1018 molecules of RB dye (in the case of 1:1 molar ratio of the dyes in synthesizing bath). This is because the process of encapsulation of rhodamine dyes does not noticeably change their quantum yield and fluorescence spectrum [22]. However, the actual measured fluorescence brightness is equal to that of 73 molecules of R6G and 1480 molecules of RB dye (see Appendix A, which presents a graphical demonstration of obtaining these numbers). The difference can be explained by FRET because presumably a part of R6G molecules served as donors for RB molecules, and consequently, the contribution of R6G molecules became smaller, while the contribution of RB molecules increased. When analyzing the second case of the 1:0.1 molar ratio of the dyes in a synthesizing bath, one can see that the FRET effect was much smaller.

It is interesting to evaluate the efficiency of FRET in a more quantitative manner. The efficiency of FRET *E* can be found using a known equation: $E=R^{6}/\left(R^{6}+r^{6}\right)$, where *R* is the Forster radius (the latter is 8.8 nm for this dye couple (see Appendix A). Assuming an even distribution of the dye molecules in the particle volume, one can estimate the average distance between the dye molecules. These calculations show that the distance between the dye molecules is within 3.2–3.5 nm (see Table 1). This gives us a FRET efficiency close to 100% inside of these Star-dots. However, one can see from the previous paragraph that the contribution of FRET is substantially different for these two types of particles. It is worth noting that a similar nontrivial behavior of FRET was observed when mixing these dyes in micron size nanoporous particles [26]. Investigation of FRET between encapsulated dye molecules is beyond the scope of this work, which is the demonstration of the ability to obtain complex spectra by mixing dyes inside particles, and the potential use of these particles for demultiplexing applications. Therefore, we defer the study of FRET to future works.

One can also see from Table 1 that the particles are rather bright. To understand the degree of brightness, one can also compare the synthesized Star-dots with the commercially available quantum dots. It certainly makes sense to compare to quantum dots of similar fluorescence spectra. Because we have two different fluorescent dyes, we should compare its brightness to two different quantum dots. The closest to R6G is QD525 and QD585 for RB [36]. Comparing the brightness of 2-dye particles with the molar ratio of R6G to RB 1:1, one can find that each to particle has the brightness of 47 QD525 and 14 QD585 when excited at 488 nm. Similarly, the brightness of one particle synthesized with the molar ratio of 1:0.1 is equivalent to the brightness of 190 QD525 and 6 QD585 when excited at 488 nm. This is why the synthesized Star-dots can be called ultrabright (brighter than a quantum dot of similar spectra [22]).

Let us now demonstrate the application of the synthesized 2-dye particles for quantitative multiplexing. Because flow cytometry is expected to be one of the immediate applications of the developed particles, we demonstrate multiplexing using a phantom system, which is similar to the fluorescence detection in flow cytometry. Specifically, the R6G/RB particles with molar ratios of 1:1 and 1:0.1 (MR1:1 and MR1:0.1 particles) were premixed in five different proportions. A confocal fluorescent microscope with 10x magnification objective and 0.25 NA was used to collect the fluorescent light from an elliptical region of 4.5 µm of lateral and 308 µm of vertical dimension (estimated using formulas of [37]). These dimensions approximately correspond to the size of a spherical cell with a radius of 9 µm, which could be used in flow cytometry. We estimate 94 Star-dot particles located within this volume.

The examples of the fluorescent spectra collected with 100% of MR1:1 and MR1:0.1 and mix of 50% + 50%, 75% + 25%, and 25% + 75% mix of MR1:1 + MR1:0.1 particles are shown in Figure 5a. The time of collection of the fluorescence signal was taken to be 10 msec, which is about the time of measurements per event in such a high resolution multiplexed flow cytometer as CytoFLEX (by Beckman Coulter, Indiana, IN, USA). The other times of collection, starting from 1 msec and up to 500 msec were also tested (the results are shown in the Appendix A). The “calibration” spectra $FL_{\mathrm{MR}1:1\;}^{\lambda^{i}}\;\mathrm{and}\;FL_{\mathrm{MR}1:0.1}^{\lambda^{i}}$ for pure MR1:1 and MR1:0.1 particles in Equation (5) were measured in the same way, but the signal was collected for the longer time of 500 msec to minimize noise (in principle, the time of the measurements of the calibration spectra can be much longer—the spectra can be collected in advance and used just for multiplexing decomposition later). These spectra are shown in Figure 5b. Although the fluorescence spectra of the dyes mixed in different proportions shown in Figure 5a may look rather similar to the human eye, the linear decomposition algorithm allows separating the contribution of those two fluorophores quite unambiguously.

Table 2 shows the percentage of the mix from the recorded fluorescent spectra as described above. Comparing to the known mixing percentage, one can estimate the error of the demultiplexing. Note that we did consider the percentages of each particle as independent parameters. Because the errors of calculation of the concentrations of MR1:1 and MR1:0.1 particles are different, we also present the average error, which is just a simple arithmetic average of individual errors for each dye.

It should be noted that we did not limit the concentration parameters by positive numbers when applying the algorithm of demultiplexing. This is the reason why we could obtain a negative value of the concentration. This was done to estimate the error of demultiplexing. For example, there was a calculated −16% in the last row of Table 2. Because the error of demultiplexing was found as the difference between the calculated demultiplexing concentration and the actual one, we could estimate the error as 16%. If we do not need to evaluate the error of demultiplexing, we would add an additional condition of positivity of the calculated concentration, which would eliminate negative values. Because the purpose of this work was to demonstrate a relatively low error of demultiplexing, the modification of the demultiplexing algorithms was left for future work.

It should be noted that there is a simple yet powerful method to improve the accuracy of demultiplexing. To do it, the minimization of the fitting error given by Equation (2) can be modified to take into account the signal-to-noise ratio of the recorded fluorescence signal. This can be done, for example, by adding weight to the minimization of the error in Equation (2):$\sum_{i=1\;}^{m}w_{i}{\left[FL_{T}^{\lambda^{i}}-\sum_{j=1}^{n}c_{j}FL_{j}^{\lambda^{i}}\right]}^{2}.\;$ A simple choice for $w_{i}$ is to assume that the noise is relatively independent of the wavelength, whereas this signal is directly proportional to the fluorescence coming from the particles. Therefore, higher fluorescence values contribute more towards the minimization of the error in Equation (2), and consequently, to the definition of the sought concentrations. Thus, $FL_{T}^{\lambda^{i}}$ can be considered as $w_{i}$. Table 3 shows the results of demultiplexing with such a weight. One can see a substantial decrease in the error of demultiplexing. It is interesting to note that the error is not a monotonic function of the percentage of mixing. This can be explained by multiple factors, including the specific difference in multiplexing spectra, the noise of the detector at different wavelengths, etc. In principle, it is possible to tune the weight function to minimize the error for each specific case. This is, however, beyond the scope of this work.

The error of demultiplexing should obviously decrease with the increase of the collection of the fluorescence signal, or with the increase of the laser power. The laser power used in the present measurements was 116 µW, which is rather modest compared to the power of available lasers. Here we demonstrate the decrease of the demultiplexing error with the increase of the time of collection of the fluorescence signal. For example, the error can be as bad as 30% for the collection time of 1 msec (no matter if it is weighted or not) and can be decreased down to a single-digit level when the collection time riches 500 msec (see the Appendix A).

## 4. Conclusions

Here we demonstrated the method of creation of ultrabright fluorescent silica nanoparticles with very complex fluorescence spectra, which can be created by simple mixing of multiple dyes in the synthesizing bath. A particularly useful feature of exciting multiple spectra with single excitation wavelengths is demonstrated. The dye molecules are mixed inside the particles by a chain FRET coupling. The mixing of up to five different dyes within each particle is demonstrated here. Furthermore, the dyes can be encapsulated in different concentrations, which makes the number of potentially distinguishable spectra practically unlimited. Using a linear decoupling algorithm, we demonstrated a reasonable ability to demultiplex a mixture of two different particles, in which the same dyes were mixed into different proportions. It was demonstrated for a volume containing about a hundred nanoparticles. We foresee the use of these particles in a number of applications, in particular in flow cytometry, in which demultiplexing may substantially increase the speed of detection.

## Figures and Tables

**Figure 1 nanomaterials-10-00905-f001:**
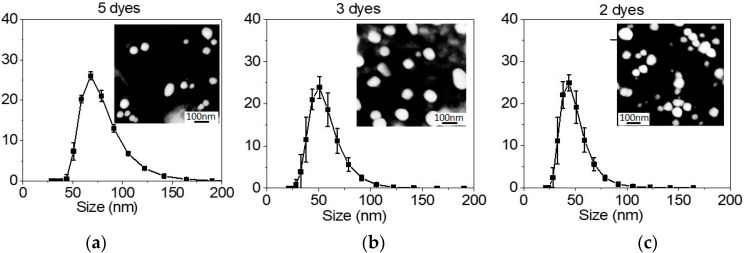
Dynamic Light Scattering (DLS) measurements of particle size distribution. The histograms of the particle sizes are shown. (**a**) Star-dots with 5 encapsulated dyes (coumarin 504 (C504), rhodamine 560 (R560), rhodamine 6G (R6G), rhodamine 640 (R640), and Nile Blue (NB)); (**b**) Star-dots with 3 encapsulated dyes (fluorescein isothiocyanate (FITC), rhodamine B (RB), and NB); (**c**) Star-dots with 2 encapsulated dyes (R6G and RB). The inserts are Atomic Force Microscopy (AFM) images of corresponding particles.

**Figure 2 nanomaterials-10-00905-f002:**
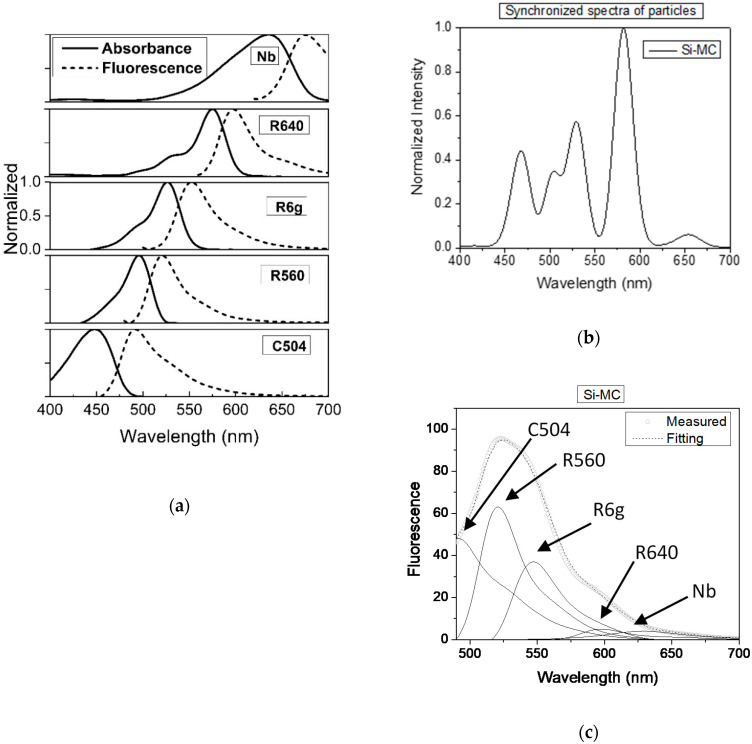
Encapsulation of C504, R560, R6G, R640, and NB dyes within each nanoparticle. (**a**) Absorption and emission (normalized) spectra of the dyes; (**b**) synchronized fluorescence spectrum of the particles scanned with the fixed difference between excitation and emission of 10 nm; (**c**) the fluorescence spectrum of Star-dot particles excited with 480 nm; contributions of individual dyes are shown with the solid lines.

**Figure 3 nanomaterials-10-00905-f003:**
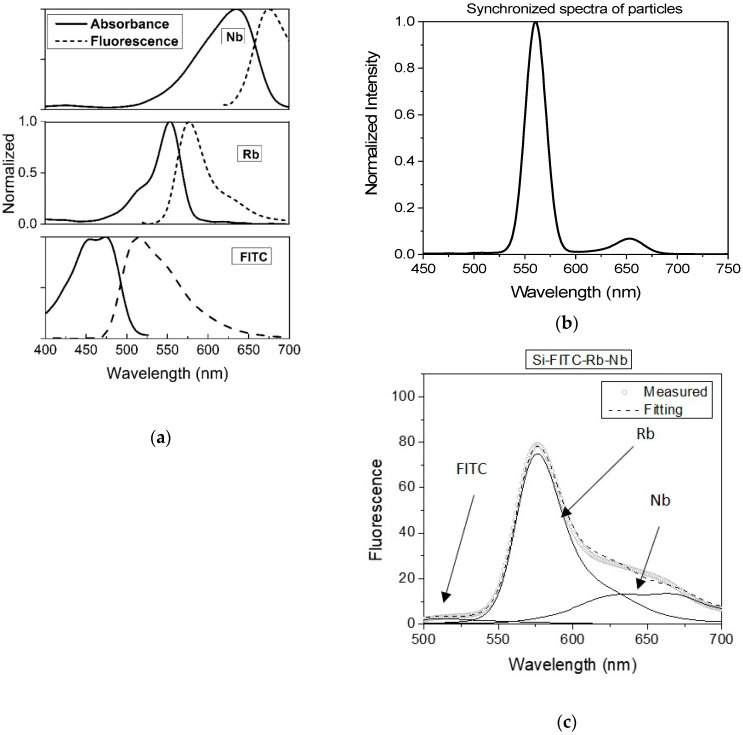
Encapsulation of FITC, RB, and NB dyes within each nanoparticle. (**a**) Absorption and emission (normalized) spectra of the dyes; (**b**) synchronized fluorescence spectrum of the particles scanned with the fixed difference between excitation and emission of 10 nm; (**c**) the fluorescence spectrum of Star dots particles excited at 488 nm; contributions of individual dyes are shown with the solid lines.

**Figure 4 nanomaterials-10-00905-f004:**
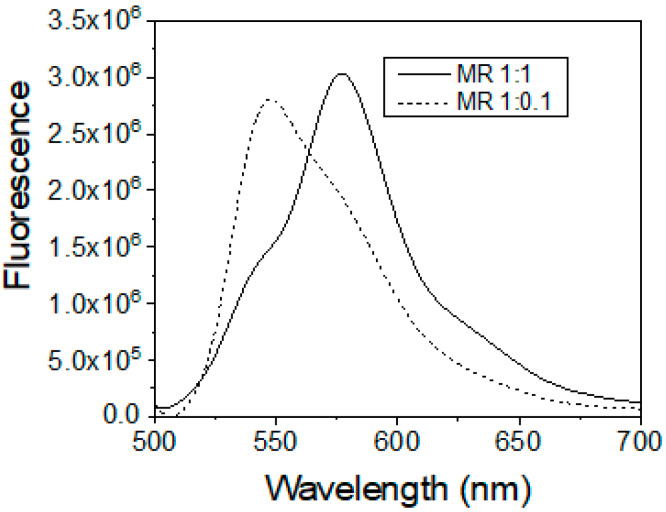
Encapsulation of R6G and RB dyes within each nanoparticle in different proportions. Fluorescence spectra of the particles for three different proportions of the dyes excited at 488 nm.

**Figure 5 nanomaterials-10-00905-f005:**
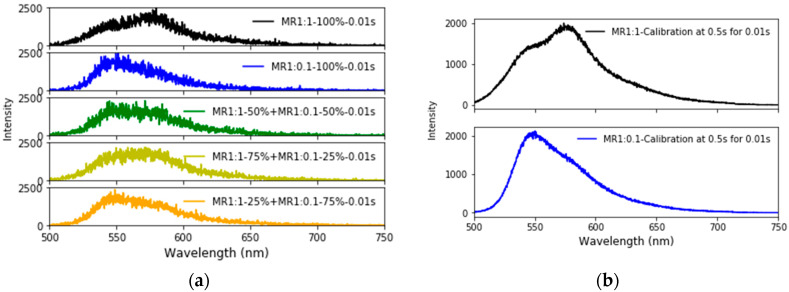
Verification of multiplexing. (**a**) Fluorescent spectra collected with 100% of MR1:1 (black) and MR1:0.1 (blue) and mix of 50% + 50% (green), 75% + 25% (yellowish green), 25% + 75% (orange) mix of MR1:1 + MR1:0.1 particles excited at 488 nm. The time of collection of the fluorescence spectrum was 10 msec. (**b**) Calibration spectra of pure MR1:1 (top) and MR1:0.1 (bottom) particles collected for 500 msec.

**Table 1 nanomaterials-10-00905-t001:** Parameters of Star-dots with encapsulated R6G and RB dyes within each nanoparticle in different proportions (shown in Figure 4).

Molar Ratio of R6G:RB in Synthesizing Bath	No. of Dye Molecules per Particle R6G + RB	The Average Distance between the Dye Molecules	FRET Efficiency	Particle Brightness in MESF ^1^ Units R6G + RB (488 nm ex.)	Particle Brightness in MESF ^1^ Units R6G + RB (max ex.) ^2^
1:1	226 + 1018	3.2	0.998	73 + 1480	27 + 205
1:0.1	317 + 592	3.5	0.996	295 + 610	108 + 85

^1^ MESF = molecules of equivalent soluble fluorochrome. ^2^ Brightness of Star-dots with respect to the maximum brightness of the reference dyes excited at 550 nm (RB) and 525 nm (R6G).

**Table 2 nanomaterials-10-00905-t002:** Demultiplexing using different mixes of MR1:1 and MR1:0.1 particles excited at 488 nm and collected for 10 msec.

Actual mix % MR1:1:MR1:0.1 Particles	Calculated mix % of MR1:1:MR1:0.1	Error in Demultiplexing MR1:1/MR1:0.1	Average Error in Demultiplexing
100:0	85:7	15/7	11
25:75	12:79	13/4	8
50:50	32:56	19/6	12
75:25	62:35	13/10	11
100:0	−16:102	16/3	10

**Table 3 nanomaterials-10-00905-t003:** Demultiplexing using different mixes of MR1:1 and MR1:0.1 particles excited at 488 nm and collected for 10 msec with a weight function proportional to the fluorescence intensity.

Actual mix % MR1:1:MR1:0.1 Particles	Calculated mix % of MR1:1:MR1:0.1	Error in Demultiplexing MR1:1/MR1:0.1	Average Error in Demultiplexing
100:0	96:−0.2	5/0.2	2
25:75	16:77	9/2	6
50:50	39:53	11/3	7
75:25	70:30	6/5	5
100:0	−9:99.9	9/0.1	5

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
