# Peer review of "Ultrabright Fluorescent Silica Nanoparticles for Multiplexed Detection"

_nanomaterials, 2020, doi:10.3390/nano10050905_

Round 1

Reviewer 1 Report

The paper is well presented and the fluorescent signal of multiple dyes are presented showing multiplexing.

It is very similar to a previous publication,

Palantavida, S., B. Peng, and I. Sokolov, Ultrabright fluorescent silica particles with a large number of complex spectra excited with a single wavelength for multiplex applications. Nanoscale, 2017. 9(15): p. 4881-4890. 

-except further dyes are combined.

Reviewer 2 Report

In this manuscript, the preparation and investigation of fluorescent silica nanoparticles encapsulated with various chromophores is presented. The authors describe their results regarding the structural and photophysical study of the particles but also on their effort to demultiplex the contribution of each chromohpre on the fluorescence spectra by following a linear decomposition method. In general, the topic is interesting and the manuscript is clearly written. However, some parts of the manuscript are highly qualitative and some conclusions are speculative. Below are my comments.

1) The authors are stringly encouraged to report fluorescence excitation spectra in order to reveal the existence of FRET mechanism among the dyes.

2) The authors state that the low fluorescence intensity of RhG, R640 in the spectrum of fig. 2c is due to the less than perfect FRET. But what is the role of the fluorescence quantum yields? Their values should be reported and discussed.

3) In the 2-dye particle, the authors say that FRET is observed since RB is not excited at 488nm. However, this statements is qualitative and does not assure the existence of FRET. Experiments should be performed on similar particles with only RB excited at 488nm and the results should be compared to those of the 2-dye system. In the current results direct excitation of the RB dye at 488nm cannot be excluded.

4) in addition to the previous comment, the authors in page 8 say that there wouldn't be a high peak of RB seen in the fluorescence spectra. This statement cannot be supported unless comparative measurements without the donor dye are performed.

5) The authors use the FRET formulation for calculating the energy transfer efficiency E to confirm that "we are dealing with FRET." However, this is not a proof for the existence of FRET. They use the formulation because they presuppose that FRET is operative.

6) The authors in the 2-dye system use several times the term "high peak of RB" or similar. This should be avoided. Whether or not the fluorescence peak of RB is of high intensity or not can only be judged with comparative measurements as discussed in comment 4.

7) The authors are asked to specify the meaning of the negative mix % in tables 2 and 3.

8) Based on the parameters given in Table 1, FRET efficiency between RhG+RB is very high. This would lead to a vanishing fluorescence of the donor dye and only the spectrum of the acceptor dye should be observed. But this is not the case. The authors maybe should reconsider their calculations.

9) Also, in the 5-dye system, increased cascade FRET among the dyes will again lead to red-shifted emission originating only from the lower energy emitter. But to my opinion this is not seen and also this should not be the goal of this work. As a general comment, if efficient FRET among all dyes is operative, then, there wouldn't be multi-emission spectra and deconvolution would not be needed. So, inefficient FRET should actually be present in order to have multiple spectra suitable for multiplexed applications. The authors should make a comment on that.  

Below are some minor comments:

1) The sentence "However, their contribution to the fluorescence spectrum excited at 480 nm decreasing." should be corrected.

2) line 303, 48nm should be corrected.

3) equation 6 should be corrected.

4) The sentence "one can find that each to die particle" is not meaningful.

5) The sentence "Therefore, higher fluorescence should be counted more towards the minimization of the error in equation (2)." is not easily understood by the reader.

The authors are asked to address those comments before reconsidering their manuscript for publication.

Round 2

Reviewer 2 Report

The authors have made an attempt to address the comments. However, there are still some unclear statements and contradictions in the manuscript that need to be clarified.

Firstly, the authors in their response letter say that the absorption spectra in figures 2a and 3a are excitation spectra. However, in the manuscript they are still called absorbance spectra. In any case such spectra can never strengthen the FRET argument. Excitation spectra should be taken in silica nanoparticles containing at least two dyes when the fluorescence is detected at the acceptor fluorescence band.

In line 284, they refer to the quantum yields (QY) for the 2-dye system saying that the QY are ~95% for both dyes. However, later (line 335) and in the SI the QY of RB is given three-times lower i.e. 0.31.

Regarding my previous comment: "Experiments should be performed on similar particles with only RB excited at 488nm and the results should be compared to those of the 2-dye system." The authors' statement that the spectrum of the dyes is not changed in the nanoscale environment, although helpful, does not unambiguously address this comment because the intensity of the spectrum is sought here and not the shape of the spectrum.

Also, regarding my previous comment: "Based on the parameters given in Table 1, FRET efficiency between RhG+RB is very high. This would lead to a vanishing fluorescence of the donor dye and only the spectrum of the acceptor dye should be observed. But this is not the case." I am not sure that the authors made a comment on that. Besides, in both concentrations of the 2-dye systems, FRET efficiency is actually 100%. So, the final outcome (i.e the fluorescence spectrum) in both cases is expected to be the same. Then, how these important differences in the fluorescence spectra (figure 4) are explained? Since in both cases, energy is almost 100% transferred to the RB dye, I would expect in both cases to detect only the fluorescence of RB. The same takes place for the 3-dye system where the fluorescence of FITC dye is not detected because of efficient FRET to the other dyes.

Round 3

Reviewer 2 Report

After reading the revised version of the manuscript and the response letter of the authors, I feel that they have made every effort to address the comments. The manuscript can be accepted for publication.